# Radiographic Evaluation of Distal Radius Fracture Healing by Time: Orthopedist versus Qualitative Assessment of Image Processing

Maria Oulianski [1,*] , Dana Avraham [1] and Omri Lubovsky [2]

1 Department of Orthopedic Surgery, Kaplan Medical Center, 1 Pasternak St., Rehovot 7661041, Israel
2 Department of Orthopedic Surgery, Barzilai Medical Center, 2 Hahistadrut St., Ashkelon 7830604, Israel
* Correspondence: oulianskim@gmail.com; Tel.: +972-54-5838783

**Abstract:** Distal radius fractures are among the most prevalent long-bone fractures in the body. Fracture healing assessment is based on clinical evaluation and radiological examinations. A lack of consensus exists regarding the radiographic criteria for fracture union. Our work examined the commonly used criteria for the assessment of fracture healing. Thirty-two patients, conservatively treated for distal radius fracture, participated in a prospective study. Enrolled patients followed protocol for 26 weeks. Four orthopedic surgeons with similar ranks were asked to evaluate three parameters of radiographic measurements for each set of radiographs, including callus formation, the presence of a fracture line, and bridging of fracture sites or sites of fracture edges in 70 radiographs. Ten patients were eligible for the study. The degree of agreement among surgeons was "good" (Cronbach's alpha): callus formation—0.8, bridging of fracture sites—0.775, blurring of fracture line gap—0.795. A timeline based on the specific week and grading system was made. Radiographic detection of callus formation was seen after the second film, between 6 and 9 weeks, and an agreement among surgeons was achieved for more than half of the patients for the blurring of the fracture gap. The radiographic healing progression of the distal radius can be detected after 6 and 9 weeks in all three parameters with good agreement between different surgeons. A timeline graph such as the one that was made in this model can be used for the follow-up of patients' fracture healing or early detection of non-union.

**Keywords:** distal radius; fracture healing; radiographs; conservative fracture treatment





## 1. Introduction

Distal radius fractures are among the most prevalent long-bone fractures in our body [1]; distal radius fractures account for one-sixth of the total fractures seen in the emergency department, yet there is no published data for the radiographic fracture healing criteria formation nor about the agreement on those criteria between different surgeons. The two leading age groups with the highest fracture rates are children between 6 and 10 years old and adults 60 to 69 [2]. Fracture healing assessment is based on clinical evaluation and radiological examinations. Clinical criteria mostly assess the pain, tenderness, and ability to bear weight and perform different activities in our daily life without pain [3]. Radiologic assessment is based mainly on X-rays, orthogonal planes, and CT scans [4]. A lack of consensus exists regarding the radiographic criteria for fracture union. Delayed assessment of non-united fracture can lead to patients' morbidity and deformities; early intervention can lead to unnecessary operations. Only 39.7 to 45.4% of surgeons use specific criteria for assessing fracture healing [3,5]. Common radiological measurements currently in use are the formation of bridging callus, the presence of fracture lines, and bridging of fracture sites or sites of fracture edges from which theoretically we can determine the stage of union [6–8]. With these radiological factors, we can assess the mechanical stability

essential for load transmission across the fracture line [9]. We hypothesized that quantitative assessment of the commonly accepted radiographic parameters for the evaluation of fracture repair allows reliable identification of fracture healing acknowledgment among all orthopedic surgeons.

In our work, we gave similar weight to all radiologic parameters, although according to Bhandari et al.'s work, three out of five orthopedic surgeons give more importance to callus over persistent fracture lines to determine union [10].

Determination of a timeline for fracture healing is critical for the diagnosis of the union. Surprisingly, there is hardly any data in the orthopedic literature that frames the formation of callus, bone bridge, or fracture gap disappearance over time. This work aimed to create a preliminary model for radiographic evaluation of fracture healing over time by building a scoring system that includes the most frequent radiological measurements for qualitative assessment of distal fracture healing.

## 2. Materials and Methods

Data were prospectively collected from patients and extracted from the medical records of the participating patients. Thirty-two elderly patients with distal radius fractures signed written informed consent for the study. Patients admitted to the medical center's emergency department (ED) with new distal radius fractures were evaluated and treated according to the current practice of fracture management and then screened for the study. Patients were selected for inclusion based on a presentation to the ED with a distal radius fracture identified using the International Classification of Diseases, 9th edition (ICD-9), diagnosis (813.42), and had conservative treatment for distal radius fracture. The included fractures were stable non-articular fractures. After the first clinical and radiographic evaluation, patients had a cast put on, and a new X-ray was taken. Eligible patients completed a set of six follow-up radiographs. Blood hematology, renal function, and calcium and phosphor levels were also gathered. One week after screening, those who signed the informed consent were re-evaluated. In the case of fracture collapse, if surgery was needed, or if blood examination was not in the normal range, the patient was disqualified from the study. Exclusion criteria included low-quality radiographs, previous surgery, patients with pins or plates in the wrist, joint diseases that affected the wrist and hand function in the injured arm, and disease that affects bone metabolism.

Enrolled patients followed the same protocol with the exact timeline (0, 1, 3, 6, 9, 12, 26 weeks) in 70 radiographs. They were then followed through the healing process of their fracture. The X-ray beam was centered 1 m above the radius fracture. Antero-posterior (AP) and lateral films (LAT) were taken, with the use of picture archiving and communication systems (PACS), edited in presentation, and then randomly arranged by a computer program.

Four orthopedic surgeons with similar ranks were chosen and blindly asked to evaluate three parameters of radiographic measurements for each set of radiographs. They were asked to grade each of the three standard parameters (callus formation, the presence of a fracture line, and bridging of fracture sites or sites of fracture edges) by a point system as presented in Table 1.

Statistical descriptive analysis of each of the parameters was conducted separately on every occasion. Statistical significance was considered at $p < 0.05$. The ANOVA test and Cronbach's alpha coefficient were evaluated for the reliability of the given data. Data analysis was performed using IBM SPSS Statistics for Windows (Version 25.0, Armonk, NY, USA). Diagrams of curves were drawn with the use of Microsoft Excel 365 (Version 2101, Microsoft Corp., Redmond, WA, USA).

**Table 1.** Grading system for the parameters under investigation.

| Points | Callus Formation | Fracture Line | Bridging of Fracture Sites |
|---|---|---|---|
| 2 | Bridging callus in 2 cortexes | Disappearance of fracture gap | Bridging in 2 cortexes |
| 1 | Bridging callus in 1 cortex | Blurring of fracture line gap | Bridging in one cortex |
| 0 | Non-bridging callus | No disappearance of fracture line/No blur detected | No bridging was seen |

## 3. Results

During the study period, 10 patients had a fulfilled radiographic follow-up. The average age of the sample group was 65.4 ± 10 years; 73% were females.

The degree of agreement of four orthopedic surgeons in all three parameters was good (Cronbach's alpha between 0.9 > alpha > 0.8): callus formation—0.8, bridging of fracture sites—0.775, blurring of fracture line gap—0.795. Statistical analysis showed no significant difference among different surgeons (Appendix A, Table A1), with the following $p$ values: callus formation—0.677, bridging of fracture sites—0.331, blurring of fracture line gap—0.238. An exception was observer B. He observed two bridging calluses in most of the examined radiographs, thus suggesting an early callus formation (60%, $p$-value—0.258). It can be seen that the distribution percentages were similar in each parameter at each recovery stage (0–2) (Table 2).

**Table 2.** Percentage of division between different parameters in all surgeons individually.

| | Bridging of Fracture Sites | | | |
|---|---|---|---|---|
| | **No cortex** | **1 cortex** | **2 cortexes** | **$p$ value** |
| Surgeon A | 17.1% | 38.4% | 44.3% | 0.001 |
| Surgeon B | 7.1% | 32.9% | 60% | 0.258 |
| Surgeon C | 27.1% | 30.0% | 42.9% | 0.001 |
| Surgeon D | 44.3% | 21.4% | 34.0% | 0.001 |
| | **Fracture line** | | | |
| | **No blurring detected** | **Blurring of fracture line gap** | **Disappearance of fracture gap** | **$p$ value** |
| Surgeon A | 8.6% | 55.7% | 35.7% | 0.001 |
| Surgeon B | 17.1% | 48.6% | 34.3% | 0.023 |
| Surgeon C | 10.0% | 64.3% | 25.7% | 0.004 |
| Surgeon D | 27.1% | 52.9% | 20.0% | 0.012 |
| | **Callus formation** | | | |
| | **No callus** | **Some callus formation** | **Calcified callus** | **$p$ value** |
| Surgeon A | 15.70% | 42.90% | 41.40% | 0.001 |
| Surgeon B | 17.10% | 45.70% | 37.10% | 0.144 |
| Surgeon C | 28.60% | 28.60% | 42.90% | 0.001 |
| Surgeon D | 48.60% | 21.40% | 30.00% | 0.001 |

In half of the radiographic films, the fracture line started to disappear between the sixth and ninth weeks. In 70% of radiographs, cortexes bridged by callus were formed, and in 69% of cases, callus was calcified at a similar period as in other disciplines. In one variable analysis, the follow-up of each parameter showed improvement over the weeks. The mean of week 1 and at week 26 represents the improvement in the results: fracture line—1.000 to 1.746, bridging of fracture sites—1.351 to 1.867, and callus formation—1.200 to 1.933 (Figure 1; Appendix A, Table A2).

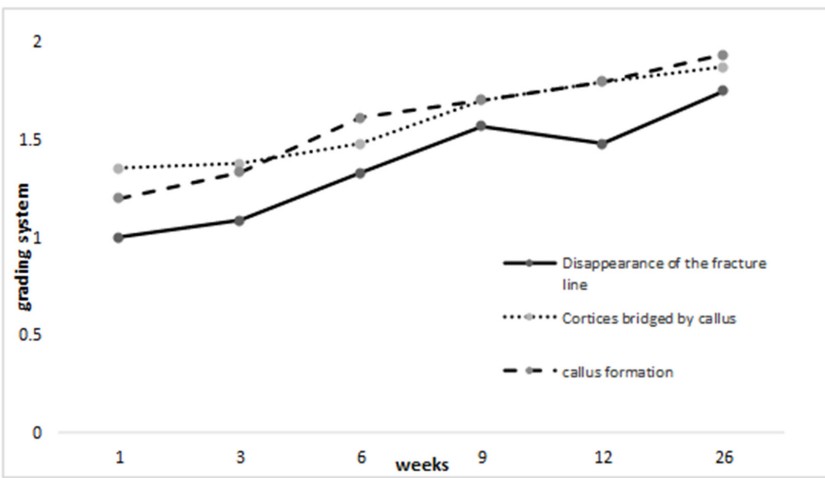

**Figure 1.** Timeline based on the grading system. Seven radiographs were obtained; only six are presented, as the first film was made at the time of casting.

## 4. Discussion

This study examined radiographic healing processes through timelines in specific time intervals for distal radius fractures that had been treated conservatively. A restricted protocol of repeated radiographs through 26 weeks enabled us to gather precise information regarding callus formation and disappearance of the fracture line without disturbance from the metallic plate in radiographs. We did not find any previous clinical study that framed the radiographic healing process of fracture over time.

An agreement was statistically achieved between the surgeons as we did not find a difference among them. A slight difference among the surgeons was in the early stages of the healing, apart from surgeon B, although it was not a significant factor. We could see a stable improvement in the results until the endpoint of all examined parameters as they approached week 26 in that they became closer to the score of 2 points which we assigned for the maximal score, suggesting the complete healing of the fracture. We observed the disappearance of the fracture line measurement on the sixth occasion. Slowing progress was seen compared to the fifth radiograph occasion. At 26 weeks, almost all parameters suggested radiologically healed fracture as they approached the maximal scoring of 2. The radiologic picture was consistent with the healing process in all three parameters. This finding corresponds to practical knowledge of the distal radius fracture healing period that assumes healing of the fracture in six to eight weeks. No universally accepted gold standard for distal radius healing is currently in use [11].

The first radiograph was taken at the time of the arrival at the hospital after achieving a reduction in a cast. Thus, the reading from the following radiographs showed better results after reduction than the actual clinical presentation. We concluded that all three of the criteria that the orthopedic surgeon examined are helpful for the continued examination of the healing process. Our study was performed on a small sample group of patients that were treated conservatively. In our view, this technic can be used for patients with external fixation as well. As the healing process is similar, some changes in the healing timeline may be seen. Radiographic films can be non-specific as different X-ray operators and machines have different approaches. The quality of the films was not always the same, and this could have influenced the orthopedic surgeon's view.

There are no published studies that built a quantitative graph of the healing process of the distal radius fracture. Our graph creates a preliminary model for the radiographic healing process. These graphs are needed on large scales to serve as a reference in evaluating the healing process of patients.

Other studies similar to ours should be conducted on other common fractures to identify specific healing timelines for each fractured bone in the body.

**Author Contributions:** M.O.—writing, original draft preparation, review and editing, D.A.—review and writing, O.L.—methodology, review and editing, project administration. All authors have read and agreed to the published version of the manuscript.

**Funding:** This research received no external funding.

**Institutional Review Board Statement:** The study was conducted in accordance with the Declaration of Helsinki. This research was reviewed by the local ethics committee of the Barzilai Medical Center and approved according to international Good Clinical Practice (GCP).

**Informed Consent Statement:** The human study protocol was approved by the Institutional Review Board of Barzilai Medical Center, protocol code 066-12.

**Data Availability Statement:** Data are available upon reasonable request to the submitting author and added as an appendix to the article.

**Conflicts of Interest:** The authors declare no conflict of interest.

## Appendix A

**Table A1.** Systematic point grading of all surgeons.

| | | N | Mean | Std. Deviation | Std. Error | 95% Confidence Interval for Mean | | Minimum | Maximum | Sig. |
|---|---|---|---|---|---|---|---|---|---|---|
| | | | | | | Lower Bound | Upper Bound | | | |
| Fracture line | A | 88 | 1.5114 | 0.50274 | 0.05359 | 1.4048 | 1.6179 | 1.00 | 2.00 | 0.677 |
| | B | 84 | 1.4167 | 0.60536 | 0.06605 | 1.2853 | 1.5480 | 0.00 | 2.00 | |
| | C | 80 | 1.4375 | 0.52395 | 0.05858 | 1.3209 | 1.5541 | 0.00 | 2.00 | |
| | D | 57 | 1.4737 | 0.50375 | 0.06672 | 1.3400 | 1.6073 | 1.00 | 2.00 | |
| | Total | 309 | 1.4595 | 0.53678 | 0.03054 | 1.3995 | 1.5196 | 0.00 | 2.00 | |
| Bridging of fracture sites | A | 88 | 1.614 | 0.5126 | 0.0546 | 1.505 | 1.722 | 0.0 | 2.0 | 0.331 |
| | B | 84 | 1.726 | 0.4994 | 0.0545 | 1.618 | 1.835 | 0.0 | 2.0 | |
| | C | 80 | 1.625 | 0.6033 | 0.0674 | 1.491 | 1.759 | 0.0 | 2.0 | |
| | D | 57 | 1.737 | 0.4828 | 0.0639 | 1.609 | 1.865 | 0.0 | 2.0 | |
| | Total | 309 | 1.670 | 0.5294 | 0.0301 | 1.611 | 1.729 | 0.0 | 2.0 | |
| Callus formation | A | 88 | 1.659 | 0.4767 | 0.0508 | 1.558 | 1.760 | 1.0 | 2.0 | 0.238 |
| | B | 84 | 1.619 | 0.4885 | 0.0533 | 1.513 | 1.725 | 1.0 | 2.0 | |
| | C | 80 | 1.750 | 0.4357 | 0.0487 | 1.653 | 1.847 | 1.0 | 2.0 | |
| | D | 57 | 1.737 | 0.4443 | 0.0588 | 1.619 | 1.855 | 1.0 | 2.0 | |
| | Total | 309 | 1.686 | 0.4648 | 0.0264 | 1.634 | 1.738 | 1.0 | 2.0 | |

**Table A2.** Follow-up through the 7 weeks in each of the parameters.

| | | Mean | Std. Deviation | Std. Error | 95% Confidence Interval for Mean | | *p* Value |
|---|---|---|---|---|---|---|---|
| | | | | | Lower Bound | Upper Bound | |
| Fracture line | week 1 | 1.2381 | 0.44 | 0.09524 | 1.0394 | 1.4368 | 0.001 |
| | week 2 | 1.0000 | 0.32444 | 0.07255 | 0.8482 | 1.1518 | |
| | week 3 | 1.0833 | 0.40825 | 0.08333 | 0.9109 | 1.2557 | |
| | week 4 | 1.3261 | 0.47396 | 0.06988 | 1.1853 | 1.4668 | |
| | week 5 | 1.5667 | 0.59280 | 0.07653 | 1.4135 | 1.7198 | |
| | week 6 | 1.4762 | 0.53452 | 0.06734 | 1.3416 | 1.6108 | |
| | week 7 | 1.7467 | 0.43785 | 0.05056 | 1.6459 | 1.8474 | |
| | Total | 1.4595 | 0.53678 | 0.03054 | 1.3995 | 1.5196 | |

**Table A2.** *Cont.*

|  |  | Mean | Std. Deviation | Std. Error | 95% Confidence Interval for Mean | | *p* Value |
|---|---|---|---|---|---|---|---|
|  |  |  |  |  | Lower Bound | Upper Bound |  |
| Bridging of fracture sites | week 1 | 1.571 | 0.5976 | 0.1304 | 1.299 | 1.843 | 0.001 |
|  | week 2 | 1.350 | 0.6708 | 0.1500 | 1.036 | 1.664 |  |
|  | week 3 | 1.375 | 0.5758 | 0.1175 | 1.132 | 1.618 |  |
|  | week 4 | 1.478 | 0.5865 | 0.0865 | 1.304 | 1.652 |  |
|  | week 5 | 1.700 | 0.5615 | 0.0725 | 1.555 | 1.845 |  |
|  | week 6 | 1.794 | 0.4079 | 0.0514 | 1.691 | 1.896 |  |
|  | week 7 | 1.867 | 0.3422 | 0.0395 | 1.788 | 1.945 |  |
|  | Total | 1.670 | 0.5294 | 0.0301 | 1.611 | 1.729 |  |
| Callus formation | week 1 | 1.476 | 0.5118 | 0.1117 | 1.243 | 1.709 | 0.001 |
|  | week 2 | 1.200 | 0.4104 | 0.0918 | 1.008 | 1.392 |  |
|  | week 3 | 1.333 | 0.4815 | 0.0983 | 1.130 | 1.537 |  |
|  | week 4 | 1.609 | 0.4934 | 0.0728 | 1.462 | 1.755 |  |
|  | week 5 | 1.700 | 0.4621 | 0.0597 | 1.581 | 1.819 |  |
|  | week 6 | 1.794 | 0.4079 | 0.0514 | 1.691 | 1.896 |  |
|  | week 7 | 1.933 | 0.2511 | 0.0290 | 1.876 | 1.991 |  |
|  | Total | 1.686 | 0.4648 | 0.0264 | 1.634 | 1.738 |  |

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
