# Peer review of "Radiographic Evaluation of Distal Radius Fracture Healing by Time: Orthopedist versus Qualitative Assessment of Image Processing"

_traumacare, doi:10.3390/traumacare2030040_

Round 1

Reviewer 1 Report

The authors present a simple pragmatic paper looking at when bony union actually occurs. I find the results reassuring that we should follow patients until at least the 12 week mark to detect union. As the authors suggest, there is a paucity of evidence objectively documenting union, so this paper serves a purpose.

Author Response

We kindly want to thank you for taking the time and reviewing our manuscript. 

Reviewer 2 Report

Thank you for the opportunity to review this paper.

I have a few issues to raise 

  • 32 patients is a very small subset in the context of distal radius fractures, 
  • What was considered the gold standard for determining union in their study? At what time did they consider the fracture united?
  • Distal radius fractures represent a very wide spectrum of injury pattern and type and healing can be influenced by many patient and injury factors, beyond the treatment methods, no description was given to patterns of fractures treated.
  •  
  • In our clinical practice X-rays are rarely viewed in isolation and the progression of X-rays are used when assessing fracture healing vs singular images ? What is the authors usual practice, and would this staging system change their clinical practice?
  • Assessment of fracture union is commonly a  clinical evaluation with commencement of range of motion (under hand therapy guidance) often commencing from any where as early as 4 weeks post injury. Further imaging is often not taken initial check xray to confirm satisfactory maintenance of position, and X-rays are only taken in the cases of patients displaying ongoing symptoms. While the patients being used in this study are typical distal radius fracture patients, those in whom it would be useful are those in which healing isn’t normal. How would the authors apply their findings to the patients, and do they feel this is a reasonable extrapolation given they excluded patients most at risk of this? 

The extrapolation that this grading could be used for assessing patients treated with devices such as ex-fixes is not entirely justifiable as fractures manageable by non-operative means vs those requiring external fixation are by that nature different entities. (Either as. A result of increased energy/fracture instability or patient factors) is there any literature on the assessment of union of distal radius fractures after ex-fix application, or similarly non-unions or osteotomies? And how does this compare to the authors findings?

Use of CT scans are commonly acceptepted method of assessing fractures that are suspected of going into non or delayed union, do the authors have any comments upon that in their study? Would they substitute this grading system for those of Ct scan?

Author Response

We kindly want to thank you for taking the time and writing a review of our manuscript. 

Q: What was considered the gold standard for determining union in their study? At what time did they consider the fracture united?

A: The gold standard for determining union was: Bridging callus in 2 cortexes, the disappearance of fracture gap, and bridging in 2 cortexes. One of the aims of our work was to examine the orthopedic surgent subjective evaluation of these parameters for fracture union.

Q: Distal radius fractures represent a very wide spectrum of injury pattern and type and healing can be influenced by many patient and injury factors, beyond the treatment methods, no description was given to patterns of fractures treated.

A: The information was added in the methods section.  The included fractures were stable non-articular fractures. Lines 69-70.

Q: In our clinical practice X-rays are rarely viewed in isolation, and the progression of X-rays are used when assessing fracture healing vs. singular images ? What is the authors usual practice, and would this staging system change their clinical practice?

A: In our practice,  during each follow-up meeting, an x-ray  ( AP and lateral ) is made . as a result, each physician can obtain and see the whole set of x-rays from the moment the patient arrives at the emergency department, including the follow-up in the outpatient clinic. 

Q:  While the patients being used in this study are typical distal radius fracture patients, those in whom it would be useful are those in which healing isn’t normal. How would the authors apply their findings to the patients, and do they feel this is a reasonable extrapolation given they excluded patients most at risk of this? 

A: This is a preliminary study examining relatively stable non-articular fractures. The aim was the formation of an evaluation system for fractures. Currently, there is insufficient data on fracture healing and callus formation in clinical practice. 

Q: The extrapolation that this grading could be used for assessing patients treated with devices such as ex-fixes is not entirely justifiable as fractures manageable by non-operative means vs those requiring external fixation are by that nature different entities. (Either as. A result of increased energy/fracture instability or patient factors) is there any literature on the assessment of union of distal radius fractures after ex-fix application, or similarly non-unions or osteotomies? And how does this compare to the authors findings?

A: In our clinical practice, external fixation for distal radius is rarely used. External fixation is used in high-energy trauma and multiple trauma settings. External fixators can be used to maintain the radial length, radial inclination effectively, and palmar tilt – in this case, non-bridging fixators should be used. Our study does not apply to those cases as the different healing process takes more time. As it was concluded in Arora J and Malik AC study (1), external fixation is good in maintaining the reduction in displaced comminuted intra-articular fractures, but the complications may be potentially devastating.

** (1) - Arora J, Malik AC. External fixation in comminuted, displaced intra-articular fractures of the distal radius: is it sufficient? Arch Orthop Trauma Surg. 2005 Oct;125(8):536-40. doi: 10.1007/s00402-005-0033-1. Epub 2005 Oct 22. PMID: 16136343.

Q: Use of CT scans are commonly acceptepted method of assessing fractures that are suspected of going into non or delayed union, do the authors have any comments upon that in their study? Would they substitute this grading system for those of Ct scan?

A: 3-dimensional callus formation process is challenging to quantify. We examined a simple cost effective available method. Using the 3D method with plain orthogonal x-rays can be extremely interesting in future studies.

Round 2

Reviewer 2 Report

Thank you for your responses, 

kind regards